# Gut Microbiota and Bacterial Translocation in the Pathogenesis of Liver Fibrosis

**DOI:** 10.3390/ijms242216502

**Published:** 2023-11-19

**Authors:** Roman Maslennikov, Elena Poluektova, Oxana Zolnikova, Alla Sedova, Anastasia Kurbatova, Yulia Shulpekova, Natyia Dzhakhaya, Svetlana Kardasheva, Maria Nadinskaia, Elena Bueverova, Vladimir Nechaev, Anna Karchevskaya, Vladimir Ivashkin

**Affiliations:** 1Department of Internal Medicine, Gastroenterology and Hepatology, Sechenov University, 119048 Moscow, Russiasedovaav@yandex.ru (A.S.); nj4@yandex.ru (N.D.); nadinskaya_m_yu@staff.sechenov.ru (M.N.); bueverova_e_l@staff.sechenov.ru (E.B.);; 2The Interregional Public Organization “Scientific Community for the Promotion of the Clinical Study of the Human Microbiome”, 119048 Moscow, Russia

**Keywords:** gut–liver axis, gut microbiota, endotoxemia, gut microbiome, cirrhosis, fibrosis, hepatitis

## Abstract

Cirrhosis is the end result of liver fibrosis in chronic liver diseases. Studying the mechanisms of its development and developing measures to slow down and regress it based on this knowledge seem to be important tasks for medicine. Currently, disorders of the gut–liver axis have great importance in the pathogenesis of cirrhosis. However, gut dysbiosis, which manifests as increased proportions in the gut microbiota of Bacilli and Proteobacteria that are capable of bacterial translocation and a decreased proportion of Clostridia that strengthen the intestinal barrier, occurs even at the pre-cirrhotic stage of chronic liver disease. This leads to the development of bacterial translocation, a process by which those microbes enter the blood of the portal vein and then the liver tissue, where they activate Kupffer cells through Toll-like receptor 4. In response, the Kupffer cells produce profibrogenic cytokines, which activate hepatic stellate cells, stimulating their transformation into myofibroblasts that produce collagen and other elements of the extracellular matrix. Blocking bacterial translocation with antibiotics, probiotics, synbiotics, and other methods could slow down the progression of liver fibrosis. This was shown in a number of animal models but requires further verification in long-term randomized controlled trials with humans.

## 1. Introduction

Cirrhosis is the end result of chronic liver disease. It is a progressive fibrosis of the liver against the background of the death of its functioning tissue and perverted regeneration, which disrupt the liver’s structure and function. According to the latest data, there are 123 million people with cirrhosis worldwide [1]. This disease contributes significantly to overall disability [2] and is very expensive to treat [3]. The main causes of cirrhosis are alcohol consumption, hepatotropic virus infection, and metabolic disorders [4]. However, not all patients with alcoholic liver disease, chronic viral hepatitis, or metabolic-associated fatty liver disease (MAFLD) develop cirrhosis [4]. It is clear that one or more additional factors contribute to the progression of chronic liver disease to cirrhosis [4]. Among them, the gut microbiota, which is a community of bacteria inhabiting the gut, has recently been highlighted [5,6,7]. The interaction of the liver with the gut and its microbiota is called the gut–liver axis [5,6,7].

The purpose of this review was to conduct a survey of published studies on the possible influence of gut microbiota pathology on liver fibrosis in chronic liver disease. Although many reviews have been published on gut–liver axis disorders in chronic liver disease, this is the first to specifically summarize the current data on the role of this axis in the development of liver fibrosis and discuss the possibility of preventing fibrosis by targeting it.

## 2. Gut Microbiota in Normal and Pathological Conditions

Normally, most of the bacteria that enter the stomach with food and from the oral cavity die there under the influence of stomach acid, which makes the stomach the most sterile organ of the gastrointestinal tract [8]. In the duodenum, the bactericidal effect of bile acids is added, but the conditions there are better than in the stomach; therefore, it contains more bacteria [9]. As contents move through the gut, the bactericidal effect decreases, and the number of bacteria increases, reaching a maximum in the large intestine. A large role is played by Bauhin’s valve, which is located between the small and large intestines and separates microbial communities whose populations differ by several orders of magnitude [10,11,12,13,14]. In addition to the bactericidal effect of stomach and bile acids and the Bauhin’s valve effect, the normal state of the gut microbiota (eubiosis) is also supported by propulsive intestinal motility, ensuring its mechanical clearance, as well as the protective properties of the intestinal barrier, which mechanically separates the contents and tissues of the intestine and produces bactericidal proteins (defensins and others) and immunoglobulins for entry into the intestinal contents [11,12,13,14]. As a result of all this, first, the small intestine contains few bacteria (fewer than 100,000 cells per mL), and second, among the intestinal bacteria, species that are beneficial to humans predominate, and conditionally pathogenic ones are minimally represented [10,11,12,13,14,15]. The interactions among the intestinal bacteria are also an important factor. For example, some bacteria produce special proteins (bactericins) that are toxic to pathogenic bacteria, while others convert primary bile acids into more bactericidal secondary ones. Still others, which consume common growth factors, compete for them with opportunistic bacteria. Many other mechanisms of interaction among the intestinal bacteria have not yet been deciphered [10,11,12,13,14,15]. The main phyla of bacteria in the human gut microbiota are Firmicutes (Bacillota), Bacteroidetes (Bacteroidota), Actinobacteria (Actinomycetota), Proteobacteria (Pseudomonadota), and Verrucomicrobiota. Normally, most of the gut microbiota comprises Firmicutes and Bacteroidetes [14].

Firmicutes are represented by two main classes: Clostridia and Bacilli. Clostridia include most of the beneficial bacteria of the gut microbiota from the *Ruminococcaceae* and *Lachnospiraceae* families. As a rule, they are obligate anaerobes and cannot exist in the presence of oxygen; therefore, they do not penetrate oxygenated living human tissues. Many of them form beneficial short-chain fatty acids (SCFAs), which are used as a source of energy by enterocytes, strengthen the intestinal barrier, and carry out regulatory functions through specific receptors [15,16,17,18,19,20]. However, there are also pathogenic species among Clostridia [21]. Among the Bacilli of the intestinal microbiota, there are many facultative anaerobes (*Streptococcaceae* and *Enterococcaceae*) that can survive and reproduce in oxygenated living human tissues [22,23,24]; thus, they can penetrate the tissues when the intestinal barrier is weakened or their population increases [22,23,24]. This penetration of bacteria from the intestinal contents into the intestinal walls and further spread throughout the human body is called bacterial translocation [22,23,24].

Bacterial translocation leads to the development of local and systemic inflammatory responses [22,23,24,25]. Chronic bacterial translocation leads to the development of chronic inflammation, which stimulates the development of fibrosis. Since blood from the intestine flows through the portal system to the liver, translocated bacteria and pro-inflammatory cytokines from the inflamed intestinal wall enter there first, stimulating the development of fibrosis in the liver [23,24,25]. Thus, bacterial translocation may be considered as one of the pathogenetic mechanisms within the gut–liver axis that stimulates fibrosis in the liver. In this regard, most gut bacteria from the Bacilli class can be considered as potential stimulants of liver fibrosis since they are substrates for bacterial translocation, while most bacteria from the Clostridia class, which strengthen the intestinal barrier, preventing bacterial translocation, and compete with harmful gut Bacilli, can be considered to protect the liver from gut-derived fibrosis.

The role of Bacteroidetes in the gut microbiota is complex and controversial. Most of these bacteria are obligate anaerobes and are not capable of bacterial translocation. However, they have endotoxins, which have rather weak activity. The literature describes controversial effects of these bacteria [26,27,28]. Changes in their abundance in the gut microbiota as well as the abundance of families included in this phylum are associated with many diseases, and the amount of these bacteria in the intestine increases in some diseases and decreases in others [26,27,28]. Therefore, Bacteroidetes can conditionally be considered neutral. Most species of Actinobacteria in the gut microbiota are *Bifidobacteria*, which are components of many probiotics and are considered beneficial [29,30].

Proteobacteria have active endotoxins and are represented in the intestinal microbiota mainly by facultative anaerobes of the *Enterobacteriaceae* family, which are capable of bacterial translocation. *Enterobacteriaceae* endotoxins can penetrate a disordered intestinal barrier, stimulating the innate immune system and leading to the development of chronic inflammation in the intestinal wall and liver without living bacteria penetrating into the tissue (molecular bacterial translocation) [31,32,33]. Therefore, unlike Bacilli, which are only capable of cellular bacterial translocation, most Proteobacteria in the human intestine are capable of both cellular and molecular bacterial translocation, which makes them the most harmful bacteria in the intestinal microbiota in terms of the potential to stimulate liver fibrosis.

Most representatives of the Verrucomicrobiota phylum in the human gut microbiota belong to *Akkermansia muciniphila*, which is considered to be a beneficial bacterium that strengthens the intestinal barrier, suppresses the growth of harmful bacteria, and has an anti-inflammatory effect [34,35,36]. It does not have an endotoxin and is an obligate anaerobe; therefore, it is not capable of bacterial translocation [34,35,36].

Normally, the majority of gut bacteria are represented by beneficial Clostridia and neutral Bacteroidetes, while harmful Bacilli and Proteobacteria constitute a minority [14]. In pathology, the composition of the gut microbiota is disrupted, which is defined as gut dysbiosis. Among the variants of gut dysbiosis, the most interesting one occurs when the number of beneficial Clostridia decreases and the numbers of harmful Bacilli and Proteobacteria increase. This weakens the protective properties of the intestinal barrier, which contributes to the development of cellular and molecular bacterial translocation into the intestinal wall and, further, into the liver, stimulating the development of fibrosis.

There is also an important role in the process for small intestinal bacterial overgrowth (SIBO), which is defined as an increase in the content of bacteria in the small intestine of more than 100,000 cells per mL [37,38,39,40]. An increase in the bacterial population of the small intestine also can lead to bacterial translocation, promoting liver fibrosis.

Therefore, gut dysbiosis and SIBO, as pathologies of the gut microbiota, can potentially stimulate liver fibrosis through the activation of chronic liver inflammation in response to cellular and molecular bacterial translocation.

## 3. Gut Microbiota and Intestinal Barrier in Cirrhosis

Patients with cirrhosis have dysbiosis, which predisposes them to the occurrence of bacterial translocation; specifically, the content of beneficial bacteria of the classes Clostridia and *Akkermansia* in the gut microbiota decreases and the levels of harmful bacteria from the Bacillus class and the Proteobacteria phylum increase (Figure 1) [41,42,43,44,45,46,47,48,49,50,51,52,53,54]. Gut dysbiosis in cirrhosis has been associated with the development of acute-on-chronic liver failure, hepatic encephalopathy, and hypoalbuminemia [47,49,52]. Patients with cirrhosis also have damage to the intestinal barrier, the exact cause of which is not yet known. Possible causes include an impaired blood supply to the intestine due to portal hypertension, a decrease in the supply of bile acids, and the gut dysbiosis itself. The more severe the cirrhosis is, the more seriously disrupted is the intestinal barrier [53,54,55,56,57,58,59,60,61,62]. Almost half of patients with cirrhosis have SIBO, the presence of which is associated with many complications (hyperdynamic circulation, ascites, minimal hepatic encephalopathy, spontaneous bacterial peritonitis, sarcopenia) [63,64].

In patients with cirrhosis, the blood levels of endotoxins, as a marker of molecular bacterial translocation, and bacterial DNA, as a marker of cellular bacterial translocation, are increased [25,65,66,67,68]. It is interesting that, even though Clostridia and Bacteroidetes continue to predominate in the gut microbiota in cirrhosis, the DNA of harmful bacteria capable of bacterial translocation from the Proteobacteria phylum and the Bacilli class predominates in the blood of these patients [67,68], which confirms our theory.

The content of endotoxins in the blood in patients with cirrhosis is more than in healthy persons; it increases with the decompensation of liver function and directly correlates with the content of *Enterobacteriaceae* in the gut microbiota [47,55,59,69] as well as the degree of damage to the intestinal barrier [55,58]. Patients with SIBO were found to have more endotoxins in their blood than those with normal amounts of bacteria in their small intestine [63]. The abundance of *Enterobacteriaceae* and bacteria from the Bacilli class in the gut microbiota in cirrhosis is directly correlated with a poor prognosis in the short, medium, and long term [52,70]. Patients with cirrhosis and SIBO also had a poorer prognosis than patients without SIBO [71]. Interestingly, half of the patients with compensated cirrhosis who had SIBO developed fatal decompensation of liver function after a few years, while patients without SIBO had an excellent long-term prognosis [71].

Therefore, in patients with cirrhosis, which is the final stage of liver fibrosis, there are changes in the composition of the gut microbiota and intestinal barrier that predispose them to the occurrence of bacterial translocation, which stimulates chronic inflammation and increased liver fibrosis even after the causative factor of chronic liver disease is eliminated (abstention, normalization of body weight, control of hepatotropic viruses).

## 4. Gut Microbiota and Intestinal Barrier in Chronic Liver Disease at the Pre-Cirrhotic Stage

The global prevalence of alcohol use disorder is 5.1% [1]. Alcohol is the leading cause of cirrhosis globally and is responsible for almost 60% of cirrhosis cases in Europe, North America, and Latin America; approximately 35% of patients with alcohol use disorder will develop various forms of alcoholic liver disease [1,72,73]. Alcohol consumption leads to the growth of Proteobacteria and Bacilli, which are capable of bacterial translocation, in the intestines and a decrease in the abundance of Clostridia, which strengthens the intestinal barrier [74,75,76,77]. Alcohol itself and alcohol-related gut dysbiosis reduce the effectiveness of the intestinal barrier, promoting bacterial translocation [76,77,78,79,80]. Individuals who consume a moderate amount of alcohol are more likely to have SIBO than non-drinkers [81]. Animal studies support that improving the intestinal barrier integrity can ameliorate alcohol-induced liver damage [82,83,84].

Similar changes (growth of Bacilli and Proteobacteria and decreased abundance of Clostridia) are observed in HCV infection [51,85,86], which presents in 0.7% of the population [1]. Endotoxemia, which indicates molecular bacterial translocation, is also observed in chronic hepatitis C [85,86,87]. In chronic HBV infection, which affects approximately 300 million people worldwide [1], a decrease in the content of Clostridia and the growth of Bacilli were observed in the gut microbiota. At the same time, the number of Proteobacteria changed little during the pre-cirrhotic stage of HBV infection compared to healthy individuals [44]. In patients with either HBV or HCV infection at the pre-cirrhotic stage, increased blood levels of biomarkers of intestinal barrier cell death and endotoxin were observed, which indicates that these patients also had bacterial translocation [88]. In addition, increased levels of other markers of bacterial translocation, namely peptidoglycans of Gram-positive bacteria and bacterial DNA, were found in the blood of patients with chronic hepatitis B [89]. We were unable to find data on the frequency of detection of SIBO in patients with HBV or HCV infection in the pre-cirrhotic stage [81].

Patients with MAFLD, which affects 25–50% of the world’s population [1], also have a compromised intestinal barrier, SIBO, and gut dysbiosis, which are associated with bacterial translocation [90,91,92].

Therefore, even at the pre-cirrhotic stage, changes in the gut microbiota and intestinal barrier in patients with the most common chronic liver diseases can lead to the occurrence of bacterial translocation. However, almost always, these changes are less pronounced than in cirrhosis; that is, they increase as liver fibrosis progresses, completing a vicious circle.

## 5. Bacterial Translocation and Liver Fibrosis: Experimental Animal Studies

The development of liver fibrosis is based on the transdifferentiation of hepatic stellate cells (HSCs) into myofibroblasts, which intensively form collagen and other components of the extracellular matrix. The main activating signals for such transformation are transforming growth factor beta (TGF-β), platelet derived growth factor (PDGF), and damage-associated molecular patterns (DAMPs) from dead hepatocytes. The main source of these pro-fibrotic cytokines is Kupffer cells, which are activated by DAMPs and pathogen-associated molecular patterns (PAMPs). PAMPs include, among others, bacterial endotoxins (lipopolysaccharide (LPS)), peptidoglycan, and DNA. This activation occurs through Toll-like receptors (TLRs), among which the reaction of TLR4 with LPS should be especially noted. TLRs are relatively recently discovered receptors that respond to conserved PAMPs, such as LPS, bacterial flagellin, bacterial wall peptidoglycan, free RNA and DNA, and others. These receptors are important components of the innate immune system since they are responsible for its primary contact with pathogens and for launching the primary response to infection. TLR4 cannot bind directly, as the LPS molecule requires a complex assembly composed of CD14 co-receptors, which facilitates the transfer of LPS to the TLR4 complex and MD-2, an adapter molecule that modulates LPS recognition. Another cofactor is LPS-binding protein, which shuttles LPS to CD14 molecules. The association of these auxiliary molecules triggers the signal, resulting in the homodimerization of TLR4 molecules and consequent signaling. Therefore, the signals that stimulate liver fibrosis are the death of liver cells of any origin, which stimulates the activation and transformation of HSCs directly or by activating Kupffer cells, and the entry into the liver of bacterial translocation products (LPS, peptidoglycan, DNA, or bacteria themselves) with the blood of the portal vein [93,94]. Interestingly, an increase in the formation of new TLR4 in response to LPS stimulation was observed in Kupffer cells [95].

In mice that had a deactivating mutation in the TLR4 gene, when the common bile duct was ligated, the degree of liver fibrosis was significantly less than in mice with normally functioning TLR4. Those mice also produced less TGF-β and collagen and had less hepatic macrophage infiltration in their livers. Similar results were obtained in chemical experimental models of simulated liver fibrosis with CCl4 and thioacetamide. Interestingly, the degree of hepatocyte damage did not differ significantly between normal and mutant mice, which provides evidence in favor of the two-hit theory (hepatocyte damage and bacterial translocation) for the development of liver fibrosis. At the same time, mice with non-functioning TLR2 did not show significantly different levels of liver fibrosis after its development was stimulated compared to standard animals [96]. Oral administration of non-absorbable antibiotics in these experimental models significantly reduced fibrosis and macrophage infiltration of the liver as well as endotoxemia, confirming that the source of LPS was the intestine [96]. In addition, it has been shown that HSCs have TLR4, and in response to LPS stimulation, their sensitivity to TGF-β increases, and they release cytokines that recruit macrophages from the blood [96]. Another study confirmed that oral rifaximin reduced liver fibrosis in this model of cirrhosis and that this effect was TLR4-dependent [97].

In experimental alcoholic liver disease, rats injected with antibodies against TLR4 had less severe liver fibrosis as well as lower levels of fibrosis biomarkers (alpha smooth muscle actin, hydroxyproline, collagen in liver tissues) than animals whose TLR4 activity was not suppressed [98]. When LPS was added to a co-culture of Kupffer cells and HSCs, hyperactivation of the HSCs occurred, but when Kupffer cells with knockout TLR4 were used, the hyperactivation was less prominent [98]. Interestingly, the signs of liver damage in experimental alcohol liver disease are preceded by the development of intestinal barrier disorders and endotoxemia [99].

Knockout of the CD14 gene (co-receptor of TLR4) in mice that underwent bile duct ligation led to less pronounced development of liver fibrosis and a smaller increase in the levels of fibrosis biomarkers, such as liver hydroxyproline and alpha-smooth muscle actin, compared with wild-type mice after this intervention. However, there was no significant difference between animal groups for biomarkers of liver damage, such as focal necrosis, biliary cell proliferation, and inflammatory cell influx [100].

The non-absorbable antibiotic rifaximin in combination with lubiprostone, which enhances intestinal motility and, thereby, reduces SIBO, was found to diminish the severity of liver fibrosis in an experimental model of MAFLD [101]. In another study, in which MAFLD was modeled by feeding rats a diet deficient in choline, increased endotoxemia, intestinal barrier disruption, and increased TLR4 (an indirect sign of their activation), TGF-β, and collagen formation in the liver were observed, leading to liver fibrosis. The use of other non-absorbable antibiotics (polymyxin B + neomycin) led to a decrease in the severity of endotoxemia, improved the condition of the intestinal barrier, increased the reduced amount of tight junction proteins in the gut epithelium, and decreased the production of TLP4, TGF-β, and collagen, accompanied by a decrease in liver fibrosis [102].

Probiotics based on *Lactobacillus rhamnosus* GG reduced liver fibrosis after ligation of the common biliary duct in mice [103]. The use of a symbiotic consisting of prebiotics (bioactive fibers of oat bran, pectin, resistant starch, and inulin) and probiotics (10^11^ colony-forming units of each of four bacteria: *Lactobacillus paracasei*, *Lactobacillus plantarum*, *Leuconostoc mesenteroides*, and *Pediococcus pentosaceus*) in mice treated for induced MAFLD with a high-fat, choline-deficient diet led to a decrease in Proteobacteria in the gut microbiota, which had been elevated on this diet, and endotoxemia, which correlated with the prevention of the development of severe liver fibrosis. Interestingly, a significant protective effect of this symbiotic against gut dysbiosis and endotoxemia developed only at week 18 of the experiment [104]. Thymoquinone, which is the major active compound derived from medicinal *Nigella sativa*, showed the ability to attenuate the expression of CD14 and TRL4. This active biological substance significantly decreased the expression of small muscle actin alpha and collagen-I in the liver and liver fibrosis in LPS-challenged cell cultures with HSCs [105]. The use of probiotics and postbiotics (formulated from killed bacteria) based on *Akkermansia muciniphila* prevented a decrease in the protective function of the intestinal barrier; decreased the expression of smooth muscle actin alpha, PDGF, TGF-β, and collagen genes; and reduced the development of liver fibrosis in a complex toxic-metabolic model of liver damage [106].

## 6. Bacterial Translocation and Liver Fibrosis: Clinical Trials with Humans

Therefore, the use of drugs targeting bacterial translocation shows promise in preventing and slowing the progression of liver fibrosis. However, we must determine the status of research on the effects of drugs targeting the gut microbiota and intestinal barrier to prevent the progression of fibrosis in people with chronic liver disease at the pre-cirrhotic stage. Such drugs approved for use in humans include probiotics, prebiotics, synbiotics, postbiotics, and antibiotics.

Short-term use (7 days) of probiotics based on *Lactobacillus subtilis* and *Streptococcus faecium* blocked the increase in blood LPS levels in alcoholic hepatitis but only in the cirrhosis subgroup. The number of *Escherichia coli* decreased in the feces of the probiotic groups. Changes in the levels of other biomarkers were not compared between the probiotic and placebo groups in this randomized controlled trial. Liver fibrosis was not assessed in this study [107]. Short-term use (7 days) of probiotics based on *Lactobacillus rhamnosus* R0011 and *Lactobacillus helveticus* R0052 increased the abundance of beneficial *Faecalibacterium prausnitzii* and decreased the amount of harmful *Escherichia coli* and serum LPS levels in alcoholic hepatitis, among other changes. Liver fibrosis was not assessed in this randomized controlled trial [108]. Other studies of short-term probiotic supplementation for alcoholic hepatitis did not evaluate the level of liver fibrosis [74,109].

The state of the intestinal barrier and the gut microbiota, and the level of endotoxemia and liver fibrosis were not assessed when studying the effects of probiotics in chronic viral hepatitis C [110]. Probiotics have not been studied in the pre-cirrhotic stage of chronic viral hepatitis B [103].

A systematic review was recently published on the effects of probiotics, prebiotics, and synbiotics in MAFLD [111]. The use of a probiotic based on *Lactobacilli, Bifidobacteria*, and *Streptococcus thermophilus* for 12 weeks led to decreased liver stiffness and Enterobacteriaceae and Bacilli levels in the gut microbiota. Other biomarkers of gut–liver axis status were not examined in this study [112]. In contrast, a multi-strain probiotic, including 14 bacteria from the genera *Bifidobacterium, Lactobacillus, Lactococcus,* and *Propionibacterium*, had no significant effect on liver stiffness after 8 weeks of use [113]. The use of another multi-strain probiotic for 12 weeks had no significant effect on liver stiffness and endotoxemia in patients with MAFLD in a study from Korea [114]. Six-month administration of a probiotic consisting of *Lactobacillus acidophilus*, *Lactobacillus casei*, *Lactobacillus lactis*, *Bifidobacterium bifidum*, *Bifidobacterium infantis*, and *Bifidobacterium longum* did not have a significant effect on liver stiffness and the expression of tight junction proteins in the epithelium of the small intestine villi [115]. Indicators of the gut–liver axis and liver fibrosis were not examined in another study of probiotics in MAFLD [116]. None of these studies assessed liver fibrosis directly from liver biopsy results. A meta-analysis of four studies of probiotics (including these drugs as part of the synbiotics) showed that these medicines still significantly reduce liver stiffness [117].

The use of the prebiotic oligofructose for 36 weeks did not significantly affect the levels of endotoxemia and liver fibrosis in metabolic-associated steatohepatitis [118]. Liver fibrosis and endotoxemia were not assessed in another study of prebiotics and MAFLD [119].

The prebiotic inulin, taken for 17 days, did not have a significant effect on markers of bacterial translocation in alcoholic liver disease [120]. There have been no published studies on the effect of prebiotics on liver fibrosis in this disease.

Patients with MAFLD who received synbiotic treatment consisting of fructo-oligosaccharides and *Bifidobacterium animalis* did not show a significant decrease in stiffness [121]. In contrast, a symbiotic containing *Lactobacillus casei*, *Lactobacillus rhamnosus*, *Streptococcus thermophilus*, *Bifidobacterium breve*, *Lactobacillus acidophilus*, *Bifidobacterium longum*, *Lactobacillus bulgaricus*, and fructo-oligosaccharide, taken for 28 weeks, significantly reduced liver stiffness in this disease [122]. Other studies did not assess the effect of synbiotics on the intestinal barrier, liver fibrosis, and endotoxemia in patients with MAFLD [123,124,125,126].

The effects of synbiotics on bacterial translocation and liver fibrosis in alcoholic liver disease and chronic viral hepatitis B and C at the pre-cirrhotic stage have not been studied in humans. The effects of postbiotics and the long-term use of antibiotics on bacterial translocation and liver fibrosis have not been assessed in chronic liver diseases at the pre-cirrhotic stage in human studies.

In contrast to the experimental models, human studies, in most cases, have not shown a positive effect of therapy targeting the gut microbiota in terms of regression of liver fibrosis in chronic liver disease at the pre-cirrhotic stage. This contradiction may be due to the fact that the natural course of most chronic human liver diseases is very long, and liver fibrosis develops several years or decades after the onset of pathogenic action. It should also be remembered that the life expectancy of experimental animals is significantly lower, and they have higher metabolic rates, which makes it possible to demonstrate the effect of various long-acting drugs. Therefore, drugs for the prevention of liver fibrosis should be taken for years, which means that clinical studies of their effectiveness should last several years. The short-term use of these drugs, as is typically practiced in clinical trials, is clearly insufficient to demonstrate a positive effect in most cases. The use of long-term probiotics in MAFLD has shown a positive effect on liver density (an indirect marker of liver fibrosis) in a meta-analysis of human studies. Therefore, the hypothesis that therapy should be aimed at preventing the development of bacterial translocation in chronic liver disease at the pre-cirrhotic stage cannot be rejected. New studies on the long-term use of drugs that affect this link in the pathogenesis of liver fibrosis in such patients are needed for final verification of this hypothesis.

## 7. Conclusions

In chronic liver diseases, gut dysbiosis occurs even during the pre-cirrhotic stage, manifesting as an increase in the proportion of harmful facultative anaerobes capable of bacterial translocation (Bacillus class and Proteobacteria phylum) and a decrease in the proportion of beneficial obligate anaerobes (Clostridia class), which strengthen the intestinal barrier, in the gut microbiota. In addition, there is disorder of the intestinal barrier and, quite often, SIBO in these diseases. All of this leads to the occurrence of bacterial translocation, one of the main manifestations of which is endotoxemia. Organisms undergoing bacterial translocation with the blood of the portal vein enter the liver tissue, where they activate Kupffer cells through TLR4. The latter, in response to activation, produce many cytokines, including profibrogenic ones, which activate HSCs and stimulate their transformation into myofibroblasts that produce collagen and other elements of the extracellular matrix (Figure 2). Blocking bacterial translocation at various stages (by the use of antibiotics, probiotics, and other agents) could possibly slow down the progression of liver fibrosis. This has been shown in a number of animal models but requires further verification in long-term randomized controlled trials with humans.

## Figures and Tables

**Figure 1 ijms-24-16502-f001:**
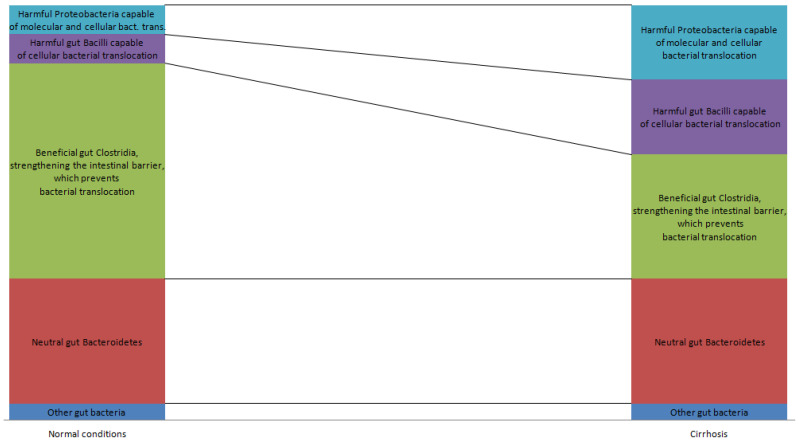
Simplified diagram of the compositional changes in gut microbiota in cirrhosis that contribute to the development of bacterial translocation.

**Figure 2 ijms-24-16502-f002:**
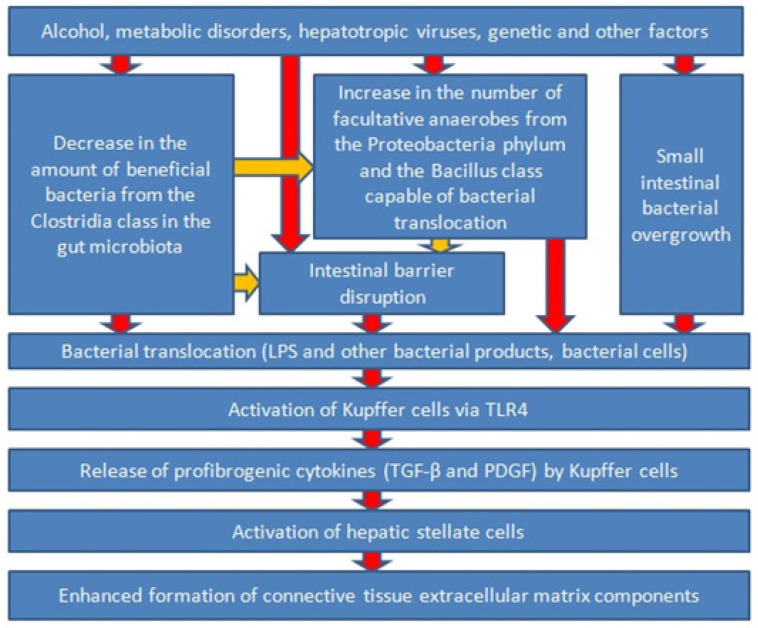
Simplified diagram showing the role of the gut–liver axis in progression of liver fibrosis in chronic liver diseases.

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
