# Peer review of "Gut Microbiota and Bacterial Translocation in the Pathogenesis of Liver Fibrosis"

_ijms, 2023, doi:10.3390/ijms242216502_

Round 1

Reviewer 1 Report

Comments and Suggestions for Authors

The authors seek to review the role of gut microbiota in the pathogenesis of liver fibrosis. The majority of the review is focused on this topic and has a few solid points. A number of reviews currently exist on this topic, or highly similar topics. While this manuscript does have some interesting points, I think the large number of reviews on this and highly similar topics highly dilutes the value of this manuscript.

Major points:

This review is considerably undercited. Many paragraphs state a number of facts with only 2-3 citations in the entire paragraph. While the latter half is better cited, considerably more citation is need in the beginning few sections.

The authors note that many microbiota have considerable controversy regarding their role but do not expand upon this aspect of things. Simply stating they are controversial just leaves them controversial. It would help to explain the different studies and why the authors come to different conclusions.

I am struggling to determine when the authors are hypothesizing and when they are summarizing. It would help to add figures and to concisely summarize points at the beginning and end of larger sections rather than break paragraphs down into individual sentences.

Some of the review simply states findings of other studies without context. There should be some sort of synthesis of new thoughts or ideas or critique/comment on the field.

The authors contract themselves some at the end:

They state in the conclusion “Blockade of bacterial translocation at various stages (use 392 of antibiotics, probiotics, and others) can slow down the progression of liver fibrosis”

and

“Therefore, drugs for the prevention 367 of liver fibrosis should be taken for years, which mean clinical studies of their effectiveness 368 should last several years. Short-term use of these drugs, as is typically practiced in clinical 369 trials, is clearly insufficient to demonstrate a positive effect in most cases. Long-term pro- 370 biotics in MAFLD has shown a positive effect on liver density (an indirect marker of liver 371 fibrosis) in the meta-analysis of human studies.”

I think the second half is likely correct. It might be more beneficial to emphasize the need for long-term clinical trials that prove the first point rather than attempt to amplify the current issues.

Comments on the Quality of English Language

Some minor improvements could be made. A thorough review by a qualified writer would be beneficial.

Author Response

The authors seek to review the role of gut microbiota in the pathogenesis of liver fibrosis. The majority of the review is focused on this topic and has a few solid points. A number of reviews currently exist on this topic, or highly similar topics. While this manuscript does have some interesting points, I think the large number of reviews on this and highly similar topics highly dilutes the value of this manuscript.

Authors' response

Dear reviewer. Thank you very much for your comment. We added the Introduction section:

“Although many reviews have been published on gut-liver axis disorders in chronic liver disease, this is the first specifically summarized the current data on the role of this axis in the development of liver fibrosis and the possibility of its prevention by targeting this axis.”

 Major points:

This review is considerably undercited. Many paragraphs state a number of facts with only 2-3 citations in the entire paragraph. While the latter half is better cited, considerably more citation is need in the beginning few sections.

Authors' response

Dear reviewer. Thank you very much for your comment.

We edited the first 3 sections of the review so that on average there were at least 4 citations per paragraph with the exception of small paragraphs, where the number of citations became 3. Thus, the first half of the review, presented in 3.5 pages, has 72 citations, and the second, consisting of 4 pages, has 58 references.

The authors note that many microbiota have considerable controversy regarding their role but do not expand upon this aspect of things. Simply stating they are controversial just leaves them controversial. It would help to explain the different studies and why the authors come to different conclusions.

Authors' response

Dear reviewer. Thank you very much for your comment.

Conflicting data concerned only the Bacteriodetes phylum. Changes in its composition, as well as in the composition of the families included in it, are associated with a large number of diseases, and in some of them there is an increase in the content of these bacteria in the intestine, and it decreases in others. A description of all these cases would take up too much space and is not relevant to the topic of our review. Interested readers can refer to the 3 cited largest reviews that are devoted to this phylum of intestinal bacteria.  One of them has the title “Bacteroides: the good, the bad, and the nitty-gritty’” which underlies the contradictory nature of this gut bacteria. We do not consider these bacteria as significant in the development of bacterial translocation in chronic liver diseases, since, according to experimental studies, they do not participate in it and do not protect against it. In our review, we tried not to overload readers with excessive information, giving them exactly as much as is required to convey and argue our ideas.

We added in the paragraph on Bacteriodetes:

“Changes in their abundance in the gut microbiota, as well as in the abundance of the families included in this phylum, are associated with a large number of diseases, and there is an increase in the amount of these bacteria in the intestine in some diseases, and a decrease in it in others [26-28].  “

I am struggling to determine when the authors are hypothesizing and when they are summarizing. It would help to add figures and to concisely summarize points at the beginning and end of larger sections rather than break paragraphs down into individual sentences.

Authors' response

Dear reviewer. Thank you very much for your comment.

Where we provide links to original research or reviews, we typically summarize their results. Where they do not this, we hypothesizes. At the end of each section, we briefly summarize the data cited and our hypotheses based on them.

Some of the review simply states findings of other studies without context. There should be some sort of synthesis of new thoughts or ideas or critique/comment on the field.

For example, in the paragraph:

“Proteobacteria have active endotoxin and are represented in the intestinal microbiota mainly by facultative anaerobes of the Enterobacteriaceae family, capable of bacterial translocation. Enterobacteriaceae endotoxin is able to penetrate a disordered intestinal barrier and, stimulating the innate immune system, lead to the development of chronic inflammation in the intestinal wall and liver without the penetration of living bacteria into human tissue (molecular bacterial translocation)[30-32]. Therefore, unlike Bacilli, which are capable only of cellular bacterial translocation, most Proteobacteria of the human intestine are capable of both cellular and molecular bacterial translocation, which makes them the most harmful bacteria of the intestinal microbiota in terms of the potential for stimulating effects on liver fibrosis.”

We first summarize the available data («Proteobacteria have active endotoxin and are represented in the intestinal microbiota mainly by facultative anaerobes of the Enterobacteriaceae family, capable of bacterial translocation. Enterobacteriaceae endotoxin is able to penetrate a disordered intestinal barrier and, stimulating the innate immune system, lead to the development of chronic inflammation in the intestinal wall and liver without the penetration of living bacteria into human tissue (molecular bacterial translocation)[30-32]«), and then, based on them and the previously cited data, we put forward our hypothesis (“Therefore, unlike Bacilli, which are capable only of cellular bacterial translocation, most Proteobacteria of the human intestine are capable of both cellular and molecular bacterial translocation, which makes them the most harmful bacteria of the intestinal microbiota in terms of the potential for stimulating effects on liver fibrosis”).

The authors contract themselves some at the end:

 They state in the conclusion “Blockade of bacterial translocation at various stages (use 392 of antibiotics, probiotics, and others) can slow down the progression of liver fibrosis”

and

“Therefore, drugs for the prevention 367 of liver fibrosis should be taken for years, which mean clinical studies of their effectiveness 368 should last several years. Short-term use of these drugs, as is typically practiced in clinical 369 trials, is clearly insufficient to demonstrate a positive effect in most cases. Long-term pro- 370 biotics in MAFLD has shown a positive effect on liver density (an indirect marker of liver 371 fibrosis) in the meta-analysis of human studies.”

I think the second half is likely correct. It might be more beneficial to emphasize the need for long-term clinical trials that prove the first point rather than attempt to amplify the current issues.

Dear reviewer. Thank you very much for your comment. We edited the Conclusion section:

“Blockade of bacterial translocation at various stages (use of antibiotics, probiotics, and others) MAY POSSIBLY slow down the progression of liver fibrosis. The latter has been shown in a number of animal models, but REQUIRES FURTHER VERIFICATION IN LONG-TERM RANDOMIZED CONTROLLED TRIALS with humans.”

Reviewer 2 Report

Comments and Suggestions for Authors

In this paper, Maslennikov et al. reviewed the relationship between gut microbiota and liver fibrosis in chronic liver disease. Despite the topic of this review is meaningful, the quality of the review needs to be improved. Here are some comments on this review:

1.        Given that the content of this paper primarily focuses on bacterial translocation, it is suggested that the title be changed to “Bacterial translocation in the pathogenesis of liver fibrosis”.

2.        The abstract should be thoroughly revised. The abstract should clarify the background, current status, significance, and future of the field.

3.        The names of bacteria in family and genus should be italicized.

4.        There are a lot of one-sentence paragraphs in the paper (e.g. lines 66-70), and the authors should revise them.

5.        Line 75 “SCFA” should be SCFAs.

6.        Lines 91-94 were confusing, please revise it.

7.        For sections 2 and 3, it is suggested to add a schematic figure, which would give a clearer picture of how bacteria change.

8.        Line 304 “drug”, could probiotics, prebiotics, synbiotics, postbiotics, and antibiotics be treated as drugs? 

Author Response

In this paper, Maslennikov et al. reviewed the relationship between gut microbiota and liver fibrosis in chronic liver disease. Despite the topic of this review is meaningful, the quality of the review needs to be improved. Here are some comments on this review:

  1. Given that the content of this paper primarily focuses on bacterial translocation, it is suggested that the title be changed to “Bacterial translocation in the pathogenesis of liver fibrosis”.

Authors' response

Dear reviewer. Thank you very much for your recommendation. We have corrected the title to "Gut microbiota and bacterial translocation in the pathogenesis of liver fibrosis"

  1. The abstract should be thoroughly revised. The abstract should clarify the background, current status, significance, and future of the field.

Dear reviewer. Thank you very much for your recommendation. We edited the abstract.

“Cirrhosis is the end result of liver fibrosis in chronic liver diseases.  Studying the mechanisms of its development and, based on this knowledge, developing measures to slow down and regress liver fibrosis seems to be an important task for medicine. Currently, disorders of the gut-liver axis are of great importance in the pathogenesis of cirrhosis. However, gut dysbiosis, which manifests in an increase in the proportion in the gut microbiota of Bacillus and Proteobacteria capable of bacterial translocation and a decrease in the proportion of Clostridia that strengthen the intestinal barrier occurs even at the pre-cirrhotic stage of chronic liver diseases. This leads to the development of bacterial translocation, those subjects with the blood of the portal vein enter the liver tissue, where they activate Kupffer cells through Toll-like receptor 4. The latter, in response to activation, produce profibrogenic cytokines, which activate the hepatic stellate cells, stimulating their transformation into myofibroblasts and their production of collagen and other elements of the extracellular matrix. Blockade of bacterial translocation at various stages can slow down the progression of liver fibrosis. The latter has been shown in a number of animal models, but requires further verification in long-term randomized controlled trials with humans.”

  1. The names of bacteria in family and genus should be italicized.

Authors' response

Dear reviewer. Thank you very much for your recommendation. The names of bacteria in family and genus were italicized.

  1. There are a lot of one-sentence paragraphs in the paper (e.g. lines 66-70), and the authors should revise them.

Authors' response

Dear reviewer. Thank you very much for your recommendation. We have corrected all one-sentence paragraphs.

  1. Line 75 “SCFA” should be SCFAs.

Authors' response

Dear reviewer, thank you very much for your recommendation. It was corrected.

  1. Lines 91-94 were confusing, please revise it.

Authors' response

Dear reviewer, thank you very much for your recommendation. It was corrected.

«In this regard, most gut bacteria from the Bacilli class may be considered as potential stimulators of liver fibrosis since they are substrates for bacterial translocation, while most bacteria from the Clostridia class, which strengthens the intestinal barrier preventing bacterial translocation and competes with harmful gut Bacilli, may be considered as liver protectors from gut-derived fibrosis.»

  1. For sections 2 and 3, it is suggested to add a schematic figure, which would give a clearer picture of how bacteria change.

Authors' response

Dear reviewer, thank you very much for your recommendation. It added Figure 1. Simplified diagram of changes in the composition of the gut microbiota in cir-rhosis, which contribute to the development of bacterial translocation

  1. Line 304 “drug”, could probiotics, prebiotics, synbiotics, postbiotics, and antibiotics be treated as drugs? 

Authors' response

Dear reviewer, thank you very much for your comment. There are several approaches to defining the word «drug». The first one is legal and considers as drug only those substances that have been registered as drugs by an authorized government agency. The second approach considers all substances that are used in medicine to be drugs, regardless of their legal registration. Indeed, the legal status of prebiotics, synbiotics, postbiotics is different in different countries, but since they are all used in medicine, we believe that they can be defined as drugs.

Round 2

Reviewer 2 Report

Comments and Suggestions for Authors

My main points and concerns have been satisfactorily addressed.